

# Worse than nothing at all: the inequality of fusions joining autosomes to the PAR and non-PAR portions of sex chromosomes

Kayla T. Wilhoit[1,2,3], Emmarie P. Alexander[4,5] and Heath Blackmon[2,4]

[1] Biomedical Sciences Program, Texas A&M University, College Station, TX, United States of America
[2] Department of Biology, Texas A&M University, College Station, TX, United States of America
[3] University Program in Genetics and Genomics, Duke University, Durham, NC, United States of America
[4] Interdisciplinary Program in Genetics and Genomics, Texas A&M University, College Station, TX, United States of America
[5] Veterinary Integrative Biosciences, Texas A&M University, College Station, TX, United States of America

## ABSTRACT

Chromosomal fusions play an integral role in genome remodeling and karyotype evolution. Fusions that join a sex chromosome to an autosome are particularly abundant across the tree of life. However, previous models on the establishment of such fusions have not accounted for the physical structure of the chromosomes. We predict a fusion joining an autosome to the pseudoautosomal region (PAR) of a sex chromosome will not remain stable, and the fusion will switch from the X to the Y chromosome each generation due to recombination. We have produced a forward-time population genetic simulation to explore the outcomes of fusions to both the PAR and non-PAR of sex chromosomes. The model can simulate the fusion of an autosome containing a sexually antagonistic locus to either the PAR or non-PAR end of a sex chromosome. Our model is diploid, two-locus, and biallelic. Our results show a clear pattern where fusions to the non-PAR are favored in the presence of sexual antagonism, whereas fusions to the PAR are disfavored in the presence of sexual antagonism.

## INTRODUCTION

Karyotypes are a fundamental aspect of genome organization that describe the number of chromosomes in a species' genome and, often, the type of sex chromosomes it possesses. Karyotypes were among the first data to be collected about genomes and predate the chromosomal theory of inheritance (*Flemming, 1882*; *Sutton, 1903*). With a century of research and many thousands of records, we might expect clear rules and generalizations about the evolution of karyotypes to have emerged. However, this is not the case. Karyotype evolution has proven incredibly recalcitrant to generalizations or rules that are applicable across large clades. Nonetheless, karyotype studies remain essential for understanding genome evolution, and as sequencing technology becomes more advanced, it will be

Corresponding author
Heath Blackmon,
coleoguy@gmail.com

increasingly possible to combine karyotype data with genomic data to develop a more nuanced understanding of the evolution of genome organization across taxa.

Chromosomal fusions and fissions are key mechanisms in the evolution of karyotype diversity across the tree of life (*Blackmon et al., 2019*). However, not all fusions are the same. Fusions joining a sex chromosome and an autosome (often referred to as neo-sex chromosomes) can have unique impacts on the fitness of an organism since the sex chromosomes segregate differentially among males and females. Fusions joining sex chromosomes and autosomes have been particularly interesting to biologists for years, because they may provide a mechanism to resolve sexual antagonism (SA). SA occurs when a gene is polymorphic for alleles benefitting one sex at the expense of the other (*e.g.*, male-beneficial genes, which are favorable for males yet detrimental to females). If an autosome has sexually antagonistic variation, a fusion with a sex chromosome could be beneficial by reducing segregation load (*Charlesworth & Charlesworth, 1980*).

Many species and clades show patterns consistent with SA variation on autosomes (*e.g.*, frequent fusions of autosomes and sex chromosomes). Multiple species within Polyneoptera have transitioned from XO to XY systems *via* an X-autosome fusion, and many beetle families have experienced sex chromosome-autosome fusions (*Blackmon & Demuth, 2014*; *Blackmon & Demuth, 2015a*; *Sylvester et al., 2020*). A recent approach to test for an excess of sex chromosome-autosome fusions showed strong support in *Habronattus* spiders but far fewer fusions than expected in *Drosophila* (*Anderson, Hjelmen & Blackmon, 2020*), with evidence suggesting that over 25% of all chromosomal fusions will be an autosome-sex chromosome fusion if the autosomal diploid count (2n) is fewer than 16, regardless of the sex chromosome system.

When we look broadly across the tree of life, we find that of 10,789 species that have been documented as male heterogametic, with over 600 having multi-XY systems that typically originate through the fusion of an autosome with an X or Y chromosome. These multi-XY systems are not restricted to little-known clades; they are present in reptiles, fish, amphibians, insects, plants, and even mammals (*Jonika et al., 2022*). For example, the artiodactyl *Gazella subgutturosa* has an XY1Y2 system that was formed by the fusion of an autosome with the ancestral X chromosome (*Tez et al., 2005*). In other mammals, such as the murine rodent *Tokudaia muenninki*, one or both sex chromosomes are thought to have undergone autosomal fusions (*Toder et al., 1997*). When compared to the other two species in the genus, which have lost their male-specific Y chromosome, the sex chromosomes of *T. muenniniki* are hypothesized to have undergone an autosomal-sex chromosome fusions to prevent the loss of the Y (*Murata et al., 2012*). In *Drosophila americana*, a putatively SA locus on a fused neo-X chromosome has fixed an inversion leading to suppressed recombination (*McAllister, 2003*). Moreover, the development of young sex chromosomes in fish and flies show signs indicating the resolution of sexual antagonism (*Kitano et al., 2009*; *Zhou & Bachtrog, 2012*). Even clades with notable stability of genome organization, such as birds, have some species with evolutionarily labile sex chromosomes. Some examples include a fusion of chr11 to the ZW chromosomes in the ancestor of order Psittaciformes (parrots) (*Huang et al., 2022*) as well as various autosome

to sex chromosome fusions in the passerine superfamily Sylvioidea (*Sigeman, Ponnikas & Hansson, 2020*).

Despite the wealth of data and interest in this area, it seems as though little attention has been given to the impact of sex chromosome structure on the fate of sex chromosome-autosome fusions. Sex chromosomes that diverge and undergo recombination suppression develop two distinctly different regions. The first is a region that maintains recombination between the X and Y, known as the pseudoautosomal region (PAR) (*Charlesworth, 1991*; *Otto et al., 2011*; *Monteiro et al., 2021*). The PAR provides a region of homology between the X and Y, which is essential for proper pairing and segregation during meiosis (*Dumont, 2017b*). Generally, species require one crossover event per chromosome or chromosome arm during meiosis (*Dumont, 2017a*). Because this recombination event must occur in the limited PAR region, the PAR's recombination rate is often orders of magnitude higher than in the rest of the genome (*Otto et al., 2011*; *Raudsepp & Chowdhary, 2015*). The second portion is often referred to as the non-recombining region, though recombination does occur along the X during female meiosis. For simplicity, we refer to this as the non-PAR. In this region, the X and Y chromosomes cease to recombine and begin to diverge and often develop radical differences in size and content. The non-PAR is also where the sex-determining gene or region is found. For the purposes of this study, the PAR of the sex chromosomes are the only regions of the chromosomes that undergo recombination, whereas the non-PAR will not recombine.

This canonical description of the evolution of chiasmatic sex chromosomes and meiosis is typical but by no means applicable to all clades. Some lineages have evolved achiasmatic meiosis in the heterogametic sex, such that meiosis occurs without chiasma formation or recombination of the autosomes or sex chromosomes (*Serrano, 1981*; *Wolf, Baumgart & Winking, 1988*; *Matioli, 1994*). Still, other lineages, like marsupials and some beetles, have asynaptic sex chromosomes (*Blackmon, Ross & Bachtrog, 2017*). In these lineages, all autosomes in both sexes experience typical chiasmatic meiosis. However, the sex chromosomes in females will experience typical chiasmatic meiosis while the sex chromosomes in males are held at a distance from one another until they are segregated to opposite poles. In marsupials, a special structure called the dense plate has evolved and is formed from synaptonemal-related proteins. During meiosis, the X and Y chromosomes adhere to opposite sides of this dense plate until it is time for them to segregate to opposite poles, at which time the dense plate dissolves, releasing the X and Y for segregation to opposite poles (*Solari & Bianchi, 1975*).

Because PARs are documented to have evolved extraordinarily high recombination rates, we hypothesize that immediately after a fusion event, the probability of a recombination event in the ancestral PAR remains high. This means an autosomal fusion will result in variable gamete types after meiosis. For instance, if a male carries a fusion of an autosome to the PAR of the X chromosome (*i.e.,* an $XY_1Y_2$ sex chromosome type), a recombination event in the ancestral PAR will result in the father producing chromosome configurations not present in the spermatogonia—an autosome to Y fusion (*i.e.,* an $X_1X_2Y$, Fig. 1). This process will occur in every generation if the PAR retains the characteristic obligatory

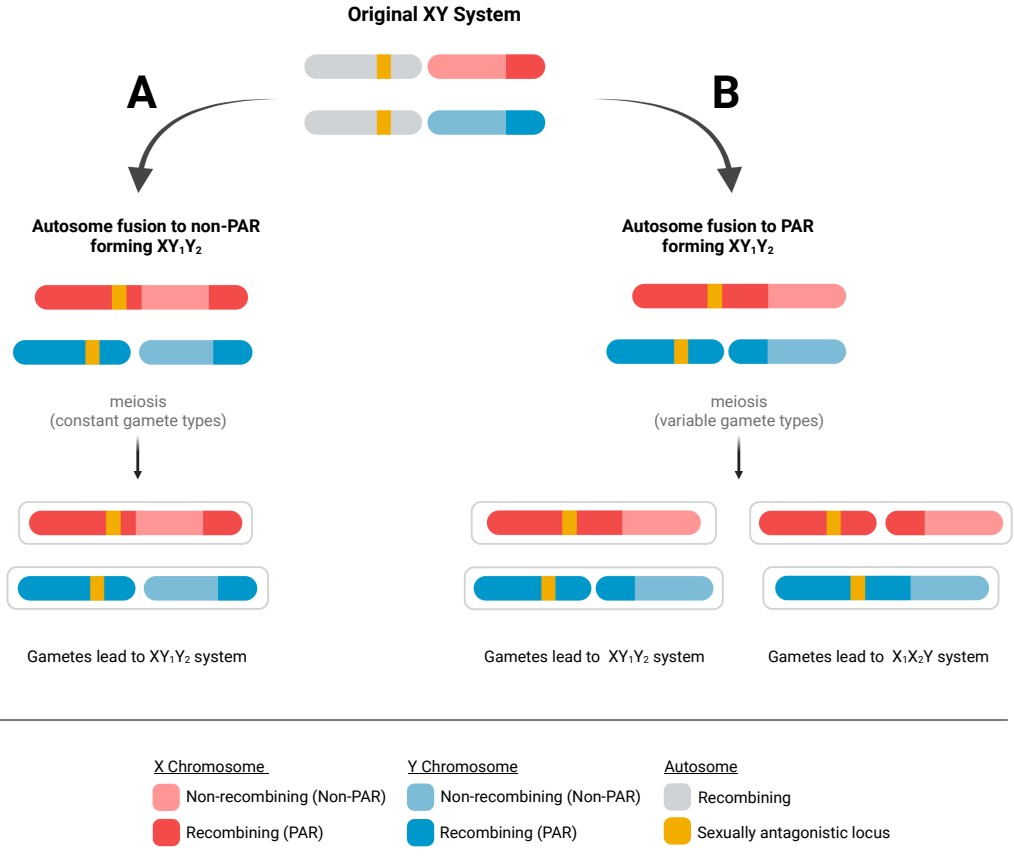

**Figure 1** **Possible fates of fusions of autosomes to the PAR and non-PAR of sex chromosomes.** When an autosome fuses to a sex chromosome, it could fuse to either (A) the non-PAR or (B) the PAR of the sex chromosome. A fusion to the non-PAR will lead to the production of stable gametes containing the same fusion as the parent. In contrast, a fusion to the PAR of a sex chromosome can produce gametes with a fusion to the Y or X chromosome after meiosis if recombination occurs in the PAR of the ancestral sex chromosome.

recombination, resulting in a variable sex chromosome system that does not allow the fusion to reduce recombination load.

To date, there is limited empirical evidence to suggest which region of sex chromosomes are more likely to fuse to an autosome—with the majority of relevant studies centered on the avian superfamily Sylvioidea. In the family Alaudidae (larks), *Alauda razae* and *Alauda arvensis* possess three autosomal fusions (chr4A, 3, and 5) to both the Z and W chromosomes. Given the presence of fusions to both sex chromosomes, a suggested scenario is one where a fusion to the PAR on one chromosome (Z, which is present in both sexes) assists in the autosomal integration to the other chromosome (W, which is female-specific) (*Sigeman et al., 2019*). Yet, in *Acrocephalus arundinaceus*—a species that possesses only one autosomal fusions (4A), linkage mapping shows evidence that the 4A fusion occurred to the non-PAR end of the chromosome (*Ponnikas et al., 2022*). This

finding is consistent with another study that suggested that the fusion point of 4A to Z was not located in the PAR in another passerine species *Ficedula albicollis* (*Smeds et al., 2014*).

For the purposes of our study, we have couched our description in terms of an XY system. However, expectations in ZW systems are thought to be essentially the same. In this work, we examine differences in the fate of autosomal fusion to either the PAR or non-PAR of the sex chromosomes. Our results show that fusions to the PAR are disfavored if the autosome has SA variation.

## MATERIALS & METHODS

Using a diploid, two-locus, biallelic model, we developed a forward-time population genetic simulation. The first locus is on the sex chromosomes and has alleles X and Y; individuals homozygous for the X allele are female, while heterozygotes are male. This locus is in the non-PAR portion of the sex chromosomes, but each sex chromosome also includes a PAR. The second locus is a sexually antagonistic locus (SAL) and is on an autosome with alleles 0 and 1; 0 is beneficial to males, while 1 is beneficial to females. The genetic architecture at this locus is described by the parameter $h$, which represents the dominance factor of the female benefit allele with possible values of 0, 0.5, and 1. Our model allows for fusions between the autosome and either the non-PAR or PAR of the sex chromosome, where fusions ultimately lead to the production of gametes with either $XY_1Y_2$ or $X_1X_2Y$ chromosomes (Fig. 1). We assume that there is an obligate recombination event in the PAR of the sex chromosomes during each male meiosis. This is an important assumption of our model, but we believe it is reasonable since failure to recombine in the PAR would be expected to lead to random segregation of the unfused sex chromosome, resulting in many aneuploid gametes. Recombination events on the autosome will occur between the fusing end of the autosome and SAL at a probability $r$, with possible values of 0.1, 0.2, and 0.4. The fitness of individuals is a function of their genotype at the SAL and their sex. We use a symmetrical model of sexual antagonism where the relative difference in possible fitness within either sex is equal (Table 1). Our selection coefficient $s$ represents the strength of selection at the SAL. There is little empirical information on the strength of selection on sexually antagonistic loci, so we evaluate selection coefficients between 0 and 1.

To simulate, we first constructed an initial population of 1,000 diploid individuals where alleles 0 and 1 are at an equal frequency on autosomes and in males and females. Under the initial conditions, no fusions were present in the population, and males and females were in equal numbers. Our simulation then cycled through generations composed of four phases: fusion mutations, selection, gametogenesis, and fertilization (Fig. 2).

We explored the fate of four fusions: (1) autosome to X PAR, (2) autosome to X non-PAR, (3) autosome to Y PAR, and (4) autosome to Y non-PAR. Fusions impacted gametes that an individual produced but had no inherent impact on fitness, and all mutations occurred prior to selection. Regardless of fusions type, during gametogenesis, recombination will occur between the two autosomes (with one autosome being fused to a sex chromosome). This recombination is obligate for proper segregation. However, this recombination event can occur between the unfused end of the chromosome and the

**Table 1  Fitness function.** Fitness is a function of the allele present at the sexually antagonistic locus and the sex of an individual. Allele 0 is beneficial to males while allele 1 is beneficial to females. The variable $h$ is the dominance factor of allele 0, and $s$ is the selection coefficient.

| Genotype | Male | Female |
|----------|------|--------|
| 00 | $1+s$ | 1 |
| 01 | $1+hs$ | $1+(1-h)s$ |
| 11 | 1 | $1+s$ |

SAL (these recombination events have no impact on gamete genotypes) or between the fused end of the former autosome and the SAL (these recombination events can impact gamete genotypes). The probability of a fusion occurring between the SAL and the fusion point is represented by the recombination rate parameter $r$. A second recombination event occurred between the already established PARs of the sex chromosomes. It is important to note that when fusions occur with the PAR, male gametogenesis often produces a fused version of the alternative chromosome (Fig. 1). Next, all individuals in the population were assigned a fitness and then gametes were drawn from parents based on the parent's fitness. During the fertilization step, eggs were chosen randomly and paired with X or Y-bearing sperm from selected parents to maintain a stable sex ratio and reconstitute the next generation of adults.

We introduced a chosen fusion in the first generation and continued the simulation for 1,000 generations. The probability of introducing a fission or fusion mutation ($\mu$) was set at 1/1000, such that the number of mutations occurring across the population in any given generation is Poisson distributed (lambda, $\lambda = 1$). For each combination of parameters (*i.e.*, $s$, $h$, $r$, and the fusion model), we performed 1,000 replicates in parallel using the doSNOW R package (*Weston, 2022*). The final genotype frequency at the end of each iteration was recorded. Increasing the number of generations was tested and did not significantly alter final genotype frequencies. For each simulation scenario, we ran our model with the strength of sexually antagonistic selection ($s$) set to zero, which allows us to observe simulation outcomes under mutation-drift equilibrium, or the condition at which the rate of new mutations arising is balanced by the rate of fixation or loss due to drift. For the remainder of the paper, model results are described in terms of deviations from mutation-drift equilibrium (hereafter MDE). All analyses were performed using R version 4.1.3 (*R Core Team, 2021*) and scripts for all analyses and figures are available in a GitHub repository: https://github.com/coleoguy/par-nonpar.

## RESULTS

The most striking pattern observed in our simulations is the difference in deviation from MDE between PAR and non-PAR fusions. In scenarios with fusions to the non-PAR of a sex chromosome, we find a consistent pattern where these fusions have increased in frequency relative to MDE (Fig. 3). In contrast, fusions to the PAR are disfavored and have a frequency below the MDE. Increasing $s$ leads to a greater deviation from MDE for both fusions—however, this effect plateaus when $s$ has values between 0.2 and 0.6, depending on scenario parameters. Below, we describe the effects on our model when varying the

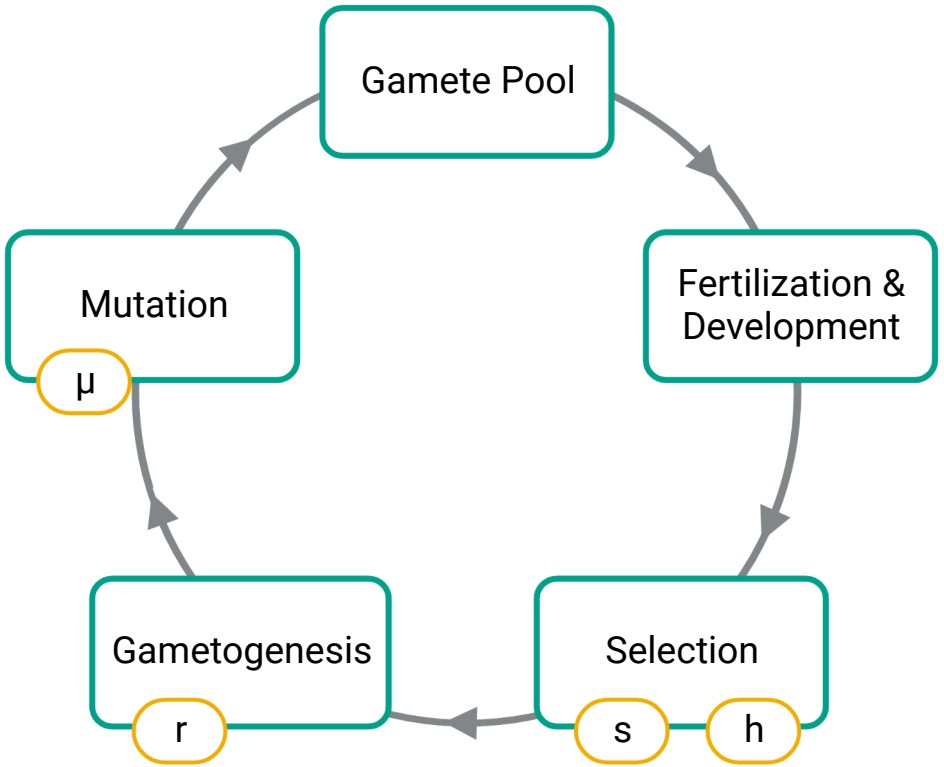

**Figure 2** **Phases of the simulation.** Large boxes indicate the stages for each generation while the small boxes show the model parameters set for each simulation. The *s* parameter is the strength of selection at the SAL, *h* is the dominance factor of the female benefit allele, *r* is the probability of a recombination event between the SAL of the autosome and the fusing end of the autosome, and μ is the probability of introducing a fission or fusion.

following parameters: the sex chromosome that is fused, the recombination distance between the fusion point and the SAL, and the dominance factor of the female benefit allele. Throughout, we discuss the average frequency across simulations for any given parameter combination. However, we note that variance across all simulations decreases as selection coefficients increase.

*Chromosome (X vs. Y):* Based on our simulations, non-PAR fusions to the Y chromosome reach marginally higher frequency than X chromosome fusions at low selection coefficients (*e.g.*, less than 0.2)—except for when the female benefit allele is recessive ($h = 0$; Fig. 3A). In such a scenario, we observe a marginally higher frequency of fusions to the X chromosome. In other scenarios, where the genetic architecture is either additive ($h = 0.5$) or dominant ($h = 1$), we observe higher frequencies of Y fusions. For instance, when the genetic architecture is additive ($h = 0.5$), the strength of selection at the SAL is low ($s = 0.2$), and the recombination distance between the fusing end of the autosome and the SAL is small ($r = 0.1$), a Y chromosome fusion event is 13% above the MDE frequency (Fig. 3B). In contrast, X chromosome fusions are 8% above MDE frequency. Depending on scenario
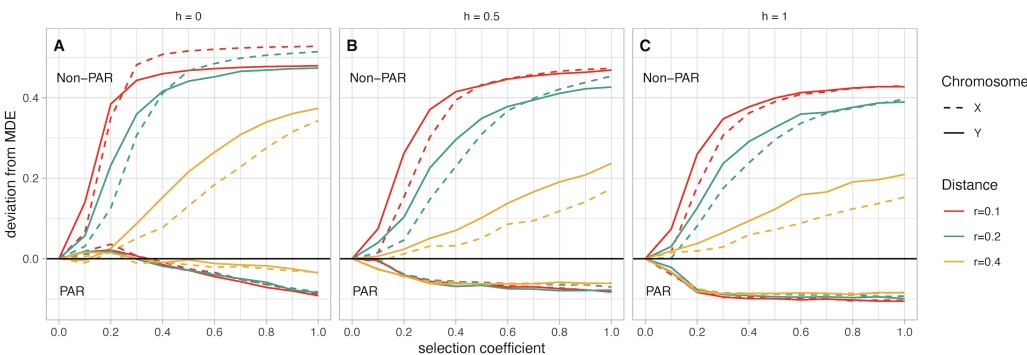

**Figure 3** **(A–C) Mean fusion frequency compared to mutation-drift equilibrium (MDE).** On the vertical axis, we plot the deviation from the MDE. The MDE is calculated as the mean fusion frequency at $s = 0$. Line color represents the value of $r$, and line pattern represents the sex chromosome which was fused. In all panels the lines that move below zero are for simulations with fusions to the PAR while those that increase above zero are for simulations with fusions to the non-PAR. Panel A contains simulations where the female benefit allele is recessive, panel B has simulations where the genetic architecture is additive, and panel C has simulations where the female benefit allele is dominant.

parameters, the frequency of X and Y fusions do converge at fixation when $s$ is between 0.3 and 0.7. Overall, the difference between X and Y fusions to the PAR is negligible.

*Recombination distance:* For non-PAR fusions, we observe a distinct pattern where increasing recombination distance reduces deviations from MDE. This pattern is consistent regardless of genetic architecture or strength of selection. The one exception is when the female benefit allele is recessive ($h = 0$), and the selection coefficient is greater than 0.7 (Fig. 3A).

*Dominance factor of the female benefit allele:* Non-PAR fusions reach fixation when the female benefit allele is recessive ($h = 0$; Fig. 3A). If the female benefit allele is additive ($h = 0.5$), fixation is reached only when $r = 0.1$ and there is strong selection, with Y fusions reaching fixation first at $s = 0.7$ and X fusions fixing at $s = 0.9$ (Fig. 3B). Models with a dominant female benefit allele ($h = 1$) never reach fixation in our simulations due to unresolved sexual antagonism (Fig. 3C). Comparatively, for PAR fusions, any increase in $h$ leads to a stronger negative deviation from the MDE, regardless of any other parameters.

## DISCUSSION

Theory predicts that fusions joining an autosome and a sex chromosome are favored when the autosome harbors sexually antagonistic variation (*Charlesworth & Charlesworth, 1980*). A fusion between an autosome and a sex chromosome can increase the linkage between an SA locus on the autosome and the sex-determining locus. Establishing linkage between loci eliminates segregation load, though some recombination load may remain. The degree of remaining recombination load is a function of the genetic architecture and the recombination rate between the SDR and SAL. This is clear in our results when we examine the average fitness in simulations with varying dominance factors. Specifically, populations achieve the highest fitness when the dominance factor of the female benefit allele is zero. A recessive female benefit allele allows the female benefit allele to fix on the

X chromosome and the male benefit allele to fix on the Y chromosome, which allows both sexes to achieve maximum fitness. In agreement with previous work *Pennell et al. (2015)*, we find that the relative difference between the fate of fusions to the X and Y is small; instead, the fate of all fusions is dominated by changes in linkage, genetic architecture, and the strength of selection.

In contrast, when the female benefit allele is dominant ($h = 1$), we see a reduction in average fitness since sexual antagonism cannot be fully resolved. In fact, any time the dominance factor of the female benefit allele is greater than zero, sexual antagonism cannot be fully resolved. When we examine simulations with a dominance factor of zero, we find that the female benefit allele at the SAL locus fixes on X chromosomes, and the male benefit allele fixes on the Y chromosome. Under this condition, every male is heterozygous and achieves peak fitness because the female benefit allele is completely recessive to the male benefit allele that is fixed on the Y chromosome. Likewise, females achieve peak fitness because all are homozygous for the female benefit allele. In contrast, as the dominance factor of the female benefit allele increases, it is less likely to fix on the X chromosome, as males that receive X chromosomes with a female benefit allele have reduced fitness. Under many conditions, this will lead to a balanced polymorphism where the X chromosome reaches an equilibrium state and both the male and female benefit alleles are maintained in the population.

It then follows that fusions to the Y should be more common than X fusions. This pattern is driven by the fact that X autosome fusions are only present in males one-third of the time. In males, the selective benefit of the fusion is felt (*i.e.,* halting recombination between alleles carried on the Y and alleles carried on the X). Thus, Y-autosome fusions are more effective in reducing the recombination that matters to the fitness of offspring (recombination among X chromosomes has no impact on offspring fitness). While fusions joining autosomes to sex chromosomes have been interpreted as a signature of resolved sexual antagonism, there are few examples where the evidence is as robust as we might wish (*Kitano et al., 2009*; *Roberts, Ser & Kocher, 2009*; *Sardell et al., 2021*). One essential piece of evidence often lacking is a measure of the fine-scale recombination rates along the newly expanded chromosome. With the advent of technologies like Clustered Regularly Interspaced Short Palindromic Repeats (CRISPR), we have the potential to move alleles trapped on Y chromosomes (due to a lack of recombination) into X chromosomes and females and to test their fitness effect directly.

Fusions between sex chromosomes and autosomes are thought to occur under one of three possible conditions. If a fusion is selectively neutral, it is expected to only reach fixation due to genetic drift. If fusions are driven through the resolution of sexual antagonism, as in our model, we would expect that only fusions to the non-PAR could reach fixation. Though we focus on sexual antagonism, we note that this is simply a special case of a coadapted gene complex and that a fused autosome could be positively selected due to interactions with a locus other than the sex-determining locus and similar results would be expected. The third option could occur under the Fragile Y hypothesis, where a recombination event within the PAR is required to properly pair and segregate sex chromosomes during meiosis. The Fragile Y hypothesis favors autosomal fusions to either the PAR or the non-PAR since it

extends the lifespan of the Y chromosome and reduces the probability of aneuploid gametes by adding new recombining sections to the sex chromosome (*Blackmon & Demuth, 2015b*; *Blackmon & Brandvain, 2017*).

To determine whether the patterns we observed in our simulations were consistent with empirical data, we searched the literature for cases where the fusion point between an autosome and a sex chromosome could be identified. We found that the Japan Sea stickleback (*Gasterosteus nipponicus*) had a fusion to the non-PAR of the Y chromosome (*Dagilis et al., 2022*).

In species of Phyllostomidae bats (*e.g.*, *Artibeus cinereus*, *Uroderma magnirostrum*) which have neo-sex chromosomes, meiotic analyses reveal a point of fusion to the non-PAR (*Noronha et al., 2010*). Moreover, evidence also suggests that the echimyid rodent *Proechimys goeldii* has a fusion to the non-PAR on the X chromosome, even in multiple karyomorphs (*Oliveira da Silva et al., 2019*). Although not an XY system, the warbler species *Acrocephalus arundinaceus* had a fusion to the non-PAR of the Z chromosome (*Ponnikas et al., 2022*). After searching through the literature, we then searched NCBI for sufficient quality genome assemblies to determine the structure of other documented sex chromosome-autosome fusions. Unfortunately, we could only access assemblies for *Muntiacus crinifrons* and *Muntiacus muntjak* (accessions GCA_020276665.1 and GCA_008782695.1). We plotted their fused X chromosomes (X+4 and X+3, respectively) against the X chromosome of the *Bos taurus* reference genome (accession GCA_002263795.3) using the software D-GENIES (*Cabanettes & Klopp, 2018*) (Fig. S1). We used the *B. taurus* reference genome as it was the most closely-related species to the *Muntiacus* spp. with no sex chromosome-autosome fusions, has an assembled X chromosome, and has documented PAR boundaries for the X chromosome (*Das, Chowdhary & Raudsepp, 2009*; *Liu et al., 2019*). We identified a fusion point to the non-PAR of the X chromosome in both *Muntiacus* species. Thus, our results, albeit with limited sample size, are consistent with predictions from our model—suggesting that autosomes are fusing to the non-PAR of sex chromosomes more often than to the PAR.

## CONCLUSIONS

Our results show a clear pattern where fusions to the non-PAR are favored in the presence of sexual antagonism, and fusions to the PAR are disfavored in the presence of sexual antagonism. This pattern primarily results from our model assuming PARs maintain high recombination rates after a fusion event. This aspect of our model is an assumption based on the fact that PARs are well documented as having among the highest recombination rates measured. These high recombination rates are likely due to sequence or epigenetic changes that will continue to control recombination rates in the short term after a fusion. Furthermore, failure to have a recombination event in the PAR region should lead to the production of aneuploidy of the unfused sex chromosome. By enforcing these high recombination rates in the PAR, we do more than reduce the linkage between the sex-determining region and the sexually antagonistic locus. If the PAR happens to be fused to an autosome, the obligate recombination event will always break apart the original

fusion and swap the former autosome to fuse to the PAR of the other sex chromosome. In this way, a fusion to the PAR is never able to resolve the recombination load due to sexual antagonism and will, in fact, serve to maximize the recombination load after every generation by reconstituting genotypes that have been selected against in the last generation. Based on the results of our model, we suggest that a fusion to the PAR of a sex chromosome under sexually antagonistic selection will be more detrimental than a state in which no fusion occurred at all. Our results show that fusions to the PAR will happen more often in the absence of sexual antagonism.

As sequencing technology continues to improve and become more affordable, the analyses that have been carried out on the *Muntiacus* species will become possible for an even greater number of organisms. This data will be key to understanding the forces that lead to common forms of genome restructuring. Our study has yielded important insights into the mechanisms underlying sex chromosome to autosome fusions. Specifically, our results suggest that if fusions join autosomes exclusively or closely to the non-PAR of sex chromosomes, then sexual antagonism may be a key force driving the fixation of these fusions. On the other hand, if fusions of autosomes to the PAR of sex chromosomes are also common, then other forces, such as the reductions in aneuploidy hypothesized under the Fragile Y hypothesis, may play a larger role in the fixation of these fusions involving sex chromosomes.

## ACKNOWLEDGEMENTS

We thank members of the H. Blackmon and W. Murphy labs at Texas A&M University for discussions of this work and feedback on earlier versions of the study.

### Funding

This work was supported by the National Institute of General Medical Sciences at the National Institutes of Health (R35GM138098). The funders had no role in study design, data collection and analysis, decision to publish, or preparation of the manuscript.

### Grant Disclosures

The following grant information was disclosed by the authors:
National Institute of General Medical Sciences at the National Institutes of Health: R35GM138098.

### Competing Interests

The authors declare there are no competing interests.

### Author Contributions

- Kayla T. Wilhoit performed the experiments, analyzed the data, prepared figures and/or tables, authored or reviewed drafts of the article, and approved the final draft.
- Emmarie P. Alexander performed the experiments, analyzed the data, prepared figures and/or tables, authored or reviewed drafts of the article, and approved the final draft.

- Heath Blackmon conceived and designed the experiments, performed the experiments, analyzed the data, prepared figures and/or tables, authored or reviewed drafts of the article, and approved the final draft.

## Data Availability

Scripts for all analyses and figures are available at GitHub and Zenodo:

- https://github.com/coleoguy/par-nonpar
- Kayla Wilhoit, & Heath Blackmon. (2024). coleoguy/par-nonpar: Pub version (Version v1). Zenodo. https://doi.org/10.5281/zenodo.12685556

## Supplemental Information

Supplemental information for this article can be found online at http://dx.doi.org/10.7717/peerj.17740#supplemental-information.

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
