# Peer review of "Worse than nothing at all: the inequality of fusions joining autosomes to the PAR and non-PAR portions of sex chromosomes"

_PeerJ, doi:10.7717/peerj.17740_

## Round 0.1 · original submission · Major Revisions

· Academic Editor

Major Revisions

I share the Reviewers' criticisms, but I encourage the Authors to resubmit. Please detail your answers in the response letter.

Reviewer 1 ·

Basic reporting

The study “Worse than nothing at all: the inequality of sex chromosome to autosome fusions” by Wilhoit, Alexander, and Blackmon applies a basic forward-in-time population genetics simulator that they developed to investigate the results of fusions of PAR and non-PAR regions of sex chromosomes. The aim of the study is interesting for understanding the evolution of these regions and my main concerns are about the simplicity of the used model and, to a minor extent, the manuscript. My general view about the study is quite positive, it could be accepted after some revisions. Please find below my specific comments.

Major comments

The results of this study are dependent on the used model, which is a very simplistic representation of reality. In general, I missed a section on the limitations and assumptions of the model, and their possible effects on the results and conclusions.

More information about the model should be included in the manuscript since the results are directly affected by the model. For example,
Does the model include birth and death events (for individuals with high or low fitness, respectively)?
Is the fitness function presented in Table 1 realistic? Could the authors indicate real examples where that function was predicted?
How many replicates were simulated under each evolutionary scenario? Is there stochasticity in these simulations? If yes, how much dispersion occurs among replicates?
Is the developed simulator validated with expectations? Are the results trustable?

More advanced forward-time simulators are available, see for example the review https://doi.org/10.1371/journal.pcbi.1002495. Why not use any of them to perform these analyses? A justification of the developed and used simulator seems needed.

The simulator is available from GitHub but there, it could include a readme file with information about how to use it (including installation, input parameters, output files, etc) would be useful for any reader interested in performing other simulations.

I think Fig 3 could include two additional graphs, in particular for h=0.25 and h=0.75, to better follow the influence of h on the deviation from MDE at different s.
Also, what is the influence of the mutation rate on the results presented in Fig 3?

Minor comments

In the introduction, I missed the recent study by Monteiro et al that provides a detailed view of the complex evolution of PARs, including linkage disequilibrium and recombination, allele frequencies, patterns of diversity, etc.
Monteiro B et al (2021) Evolutionary dynamics of the human pseudoautosomal regions. PLOS Genetics 17(4): e1009532. https://doi.org/10.1371/journal.pgen.1009532

The section Results seems too vague, it sounds almost like telegraph messages. I suggest expanding it with more details and being more fluent and connected, this improvement could be useful for readers.

The sentence “Thus, we could confirm, albeit with a limited sample size, that autosomes are fusing to the non-PARs of sex chromosomes more often than to the PAR of sex chromosomes”. A single case is not sufficient to support that affirmation. I suggest rewriting the sentence to be more cautious.

Experimental design

All the comments are presented in the section "Basic reporting"

Validity of the findings

All the comments are presented in the section "Basic reporting"

Additional comments

All the comments are presented in the section "Basic reporting"

Reviewer 2 ·

Basic reporting

This manuscript explores how the degree of linkage to the sex locus can affect the fate of a chromosome fusion in a system with sexually antagonistic polymorphism. The benefit of linking sex-determining loci and sex-antagonistic loci has been well studied, so this work focuses on understanding how this benefit is reduced in cases when the chromosomal mutation only partially links the two genetic loci (i.e a fusion to the pseudo-autosomal region, or PAR). This brief paper is generally well written and structured. The key conclusions are clear and supported by the results. Some aspects of the methods and patterns observed are unclear, however. I describe those salient issues below.

Clarify the recombination parameter, in the text (L148) and in Figures 1 and 3. Intuitively, r=0 for the fusion to the non-recombining region. However, it seems like that is not the case (according to diagram in Figure 1 and results in Fig3). For instance, what is the difference between r=0.1 in the PAR vs non-PAR simulations?

What were the stopping criteria for a run of the simulation? This is particularly important because the main results are presented in reference to a mutation-drift equilibrium, but it's not clear how the equilibrium was achieved.

(L233-241): Pennell et al (2015) found that sexual antagonism is unlikely to contribute to the observed bias in A-to-autosome fusions. Their results should be discussed in the context of the speciulation presented here.

Experimental design

No comment

Validity of the findings

It is unclear how it is possible that the fusion to the PAR (with, say, r =0.4) is disfavored (seen, e.g. in Fig 3 and mentioned in L.175). As the authors note, some linkage to sex should be beneficial for sex-antagonistic loci. If this linkage is loose (as for the values of r explored), the benefit should be very small. However, I do not see why the fusion does worse than an unfused (r=0.5) karyotype in a system with sex-antagonism.

Additional comments

Consider using the more common terms "non-recombining region (NRR)" or "sex-determining region (SDR)" instead of "non-pseudoautosomal region". Also, revise for consistency throughout (e.g. "SDR" is used in the Discussion, L.213). Related, clarify when mentions of "sex chromosomes" refer to non-recombining chromosomes (e.g, in the Introduction and when referencing previous work that treated sex chromosomes as non-recombining).

Consider revising the title. I do not see how the "inequality" is addressed by the results highlighted in the abstract, discussion, and conclusions.

Reviewer 3 ·

Basic reporting

Basic Reporting:

The authors use population genetic models to determine if sex-chromosome to autosome fusions are more likely to fix if the autosome fixes to the PAR or the nonrecombining region. Their simulations predict that fusions with the PAR are unstable, as they should switch between the X and Y chromosomes during recombination at each generation. The model considers two loci – the nonrecombining X/Y (effectively as a single locus) and a sexually antagonistic (SA) locus on the autosome. They vary dominance in both the X/Y locus and the SA locus.

I appreciate that the authors wrote this manuscript quite clearly and simply, so that even a non-theoretician (like myself) could mostly understand the simulations.

Data/code are available as required on github.

Experimental design

The research question is well-defined and meaningful, and the authors identify a knowledge gap. Methods are described in sufficient detail to replicate.

Validity of the findings

Conclusions are stated clearly, and linked to the research question, and mostly limited to supporting the results, though as my comments indicate below, in some places some context/clarification is helpful.

Additional comments

1. One of the assumptions of the model is that even after the fusion event, recombination occurs in the PAR. I’m not fully convinced we can say this with confidence, as it is known (at least over longer timescales) that fusions often alter the recombination landscape (Yoshida et al. 2023, Nature E&E for example). Do the authors have a reference for this assertion – that in the immediate aftermath of a fusion the ancestral recombination patterns are maintained? It seems like a nontrivial assumption that has important implications for the utility/applicability of this model.

2. One thing that I kept thinking about when reading the manuscript was “what do the empirical data say.” found only a handful of examples where it is known if it is the PAR or nonPAR that is fused to the autosome. I think this could be included as part of the motivation for the study, as that we don’t have good data yet makes this study more relevant. I realize that the authors only model XY systems, but as they note, the dynamics should apply in ZW as well. Therefore, I think it is worth looking for some ZW neo-sex chromosomes in where this information is known (Sylvioidea neo-sex chromosomes studied by the Hanssen group, potentially).

3. It’s unclear to me why the authors chose certain parameter values for the model – for example: r = 0.1, 0.2, 0.4 – and how they define “low” selection at s=0.2. This latter value does not seem “low” to me- perhaps I am missing something? Could the authors elaborate on this more?

4. I’m not convinced by the argument made in lines 247-255 that these are the only situations you expect a fusion between a sex chromosome and an autosome. Presumably fusions occur between sex chromosomes and autosomes simply because of linking two (or more) co-adapted alleles at epistatistically interacting loci - even in the absence of sexual antagonism?

5. It would be useful to refer to Figure 1 in the description of the methods.

---

## Round 0.2 · accepted · Accept

· Academic Editor

Accept

Dear Dr. Blackmom,

Thank you for addressing the concerns raised by the reviewers. Both reviewers agree that your paper can now be accepted for publication in PeerJ.

Sincerely,
Dr. Shaw Badenhorst

Reviewer 1 ·

Basic reporting

no comment

Experimental design

no comment

Validity of the findings

no comment

Additional comments

I found that this new version is more clear and detailed. I do not have additional comments.

Reviewer 3 ·

Basic reporting

No comment. Reporting is great.

Experimental design

No comment. Experimental design is great.

Validity of the findings

No comment. Findings are valid.

Additional comments

Thank you for your hard work and revisions. I enjoyed reading the manuscript and would be happy to see it published in its current form.